# Biomarkers Regulated by Lipid-Soluble Vitamins in Glioblastoma

**DOI:** 10.3390/nu14142873

**Published:** 2022-07-13

**Authors:** Dina El-Rabie Osman, Brandon Wee Siang Phon, Muhamad Noor Alfarizal Kamarudin, Stephen Navendran Ponnampalam, Ammu Kutty Radhakrishnan, Saatheeyavaane Bhuvanendran

**Affiliations:** 1Jeffrey Cheah School of Medicine and Health Sciences, Monash University Malaysia, Bandar Sunway 47500, Malaysia; dina.osman@monash.edu (D.E.-R.O.); brandon.phon@monash.edu (B.W.S.P.); muhamadnoor.alfarizal@monash.edu (M.N.A.K.); ammu.radhakrishnan@monash.edu (A.K.R.); 2Hospital Kuala Lumpur, Jalan Pahang, Kuala Lumpur 50586, Malaysia; ponnams99@yahoo.com

**Keywords:** biomarkers, glioblastoma cell lines, lipid-soluble vitamins, molecular mechanisms

## Abstract

Glioblastoma (GBM), a highly lethal form of adult malignant gliomas with little clinical advancement, raises the need for alternative therapeutic approaches. Lipid-soluble vitamins have gained attention in malignant brain tumors owing to their pleiotropic properties and their anti-cancer potential have been reported in a number of human GBM cell lines. The aim of this paper is to systematically review and describe the roles of various biomarkers regulated by lipid-soluble vitamins, such as vitamins A, D, E, and K, in the pathophysiology of GBM. Briefly, research articles published between 2005 and 2021 were systematically searched and selected from five databases (Scopus, PubMed, Ovid MEDLINE, EMBASE via Ovid, and Web of Science) based on the study’s inclusion and exclusion criteria. In addition, a number of hand-searched research articles identified from Google Scholar were also included for the analysis. A total of 40 differentially expressed biomarkers were identified from the 19 eligible studies. The results from the analysis suggest that retinoids activate cell differentiation and suppress the biomarkers responsible for stemness in human GBM cells. Vitamin D appears to preferentially modulate several cell cycle biomarkers, while vitamin E derivatives seem to predominantly modulate biomarkers related to apoptosis. However, vitamin K1 did not appear to induce any significant changes to the Raf/MEK/ERK signaling or apoptotic pathways in human GBM cell lines. From the systematic analysis, 12 biomarkers were identified that may be of interest for further studies, as these were modulated by one or two of these lipid-soluble vitamins.

## 1. Introduction

Gliomas are the predominant form of primary intracranial tumors that accounts for 80% of brain neoplasms [1]. Glioblastoma (GBM) is a highly lethal form of malignant glioma frequently diagnosed in adults [2]. Despite advancements in diagnostic techniques and early detection, GBM has a poor prognosis with a median survival of about 15 months. The main reason for the poor prognosis is the heterogeneity and invasive nature of GBM [3]. Lately, the World Health Organization (WHO) had integrated some genotypic parameters to its classical phenotypic scale of grading adult malignant gliomas. Grade IV GBM with the positive isocitrate dehydrogenase marker “(IDH)-wildtype” represent 90% of the total cases of malignant gliomas [4]. The GBM IDH-wildtype has a minimum of one aberrant genetic marker in one or more genes, such as the amplification of the *epidermal growth factor receptor (EGFR)* gene, loss of chromosome 10q, mutations in the *phosphatase* and *tensin homolog (PTEN)* gene, and/or deletion of the *cyclin-dependent kinase inhibitor 2A (p16INK4a)* gene [3,5].

In spite of the hostile multimodal standard treatment approach, therapy against GBM is complicated due to the progressive and heterogenic nature of the tumor [6]. At present, the treatment strategy for GBM is to perform maximal surgical resection followed by radiotherapy and concomitant administration of temozolomide (TMZ), a chemotherapeutic drug [7]. TMZ is an alkylating agent that exerts cytotoxicity by inducing methylation at the O^6^ position of the guanine base of the DNA, inducing DNA damage through thymine pairing instead of cytosine, ultimately leading to cell cycle arrest and apoptosis [8]. Despite its lipophilic nature, resistance to TMZ is observed in GBM cells and cancers that express high levels of O^6^-methylguanine-DNA methyltransferase (MGMT), which is a DNA repair enzyme that plays an important role in promoting chemoresistance to alkylating agents [8]. Thus, the limitations of conventional treatment increase the need for newer therapeutic approaches.

Natural compounds, such as vitamins, are gaining significant interest as chemo-preventive agents in cancer treatment owing to their pleiotropic properties. Vitamins are a group of essential organic nutrients that play pivotal roles in various body functions and metabolic pathways and take part in diverse cellular functions. There is growing empirical evidence, which show that vitamins have positive effects on various diseases, in particular their role in cancer prevention and treatment [9]. In general, vitamins A, D, E, and K are classified as lipid-soluble, while vitamins B complex and C are water-soluble. These vitamins can be obtained through the diet as human cells are incapable of synthesizing these micronutrients to meet the body’s needs [10]. The anticancer properties of lipid-soluble vitamins have been evaluated in GBM cells as these vitamins are able to cross the blood–brain barrier (BBB) to access these tumors.

This paper is a systematic descriptive review paper that gathered research articles published in the past 15 years on the effects of four lipid-soluble vitamins (A, D, E, and K) on GBM cells with the aim to elucidate how these micronutrients may exert anticancer effects on GBM cells and to understand the role drawn by the differentially regulated biomarkers in GBM cells and their clinical relevance.

## 2. Materials and Methods

### 2.1. Search Strategy

The PRISMA guidelines [11] were used to conduct a comprehensive systematic search was commenced using five databases (Scopus, PubMed, Ovid MEDLINE, EMBASE via Ovid, and Web of Science). In addition, suitable research articles were also manually searched using Google Scholar. The following keywords and Boolean connectors were used in the database search strategy: “Glioblastoma” OR “Glioblastoma multiforme” OR stage IV brain tumour* OR “GBM” AND vitamin* OR retinol* OR calciferol* OR tocopherol* OR tocotrienol* OR phytonadione* OR “vitamin K” OR “vitamin E” OR “vitamin D” OR “vitamin A” AND “Proteogenomic*” OR “Protein*” OR “protein expression” OR “Proteomics*” OR “protein folding” OR “Peptides*” OR “Amino acids” OR “gene protein” OR “gene expression” OR gene* OR “gene upregulation” OR “gene amplification” OR “gene dysregulation” OR “transcription factor.” Only English articles published from 2005 to June 2021 were included.

### 2.2. Study Selection

Covidence, an online systematic review software (www.covidence.org, accessed on 22 June 2021), was used to manage and select the research articles identified in the various databases [12]. All articles that came up in the search were initially uploaded to the Endnote reference management software [13], where duplicate articles were removed. Then, the articles were exported into Covidence, where duplicate articles were automatically detected and eliminated by the system. The research papers were first screened for selection using title and abstract screening, and this was followed by a full text review to identify eligible research articles based on the inclusion and exclusion criteria of this study. In both the screening steps, two independent researchers (D.E.O. and M.N.A.K.) independently reviewed and selected suitable research articles. A third independent researcher (S.B.) resolved any conflicts in both screening steps. The inclusion criteria were that the selected papers must be original research articles that evaluated the effects of natural lipid-soluble vitamins (A, D, E, and K) on gene or protein expression in human GBM cell lines. Some of the exclusion criteria included: research articles not written in the English language; non-research articles (e.g., review articles, case reports, perspective, and systematic/scoping reviews); research using animal studies or animal cell lines; and original articles that did not report on the regulation of gene(s) or protein(s) as an outcome.

### 2.3. Biomarkers Modulated by Lipid-Soluble Vitamins in Glioblastoma Cells

The role of several biomarkers regulated by lipid-soluble vitamins in human GBM cells were analyzed and short-listed. Replicable biomarkers, i.e., biomarkers that were identified to be reported in two or more research papers, were short-listed and analyzed to identify putative anticancer pathways that were regulated by vitamins A, D, E, and K in human GBM cells.

### 2.4. Survival Analysis of Biomarkers

Replicable biomarkers correlating with the survival of patients with GBM were analyzed using The Cancer Genome Atlas (TCGA) dataset. The gene expression profiles of GBM patients of cohort’s level 3 were retrieved from UCSC Xena [14]. The Agilent G4502 microarray log2 transformed data retrieved for a total of 585 GBM patients from the UCSC Xena.cBioPortal were used to acquire clinical data on the patients, such as overall survival, survival status, and the type of therapy received [15]. Patients who lacked information on their overall survival and survival status were excluded from the study. To ensure that only data from GBM patients who had received at least one of treatment were included in the analysis, only patients who lived longer than 30 days were selected for this study, which resulted in a total of 509 patients. Based on the median expression patterns, the patients were sorted into high and low expression groups. The overall survival of patients was compared using log-rank tests in RStudio, (Integrated Development for R, version 4.1.1, PBC, Boston, MA, USA) with the ‘survival’ and ‘survminer’ packages. A *p*-value of <0.05 implied a statistically significant result.

## 3. Results

### 3.1. Literature Search Results

A total of 2354 articles were initially secured (with no filters implied) from the five databases: 556 articles were obtained from Scopus, 292 articles were from PubMed, 425 articles were from Ovid MEDLINE, 790 articles were from Embase via Ovid, 236 were found through Web of Science, and 55 articles were picked from the manual search on Google Scholar (Figure 1). A total of 1048 articles identified as being duplicates were automatically detected and eliminated by Covidence’s operating system. When the remaining 1036 articles were reviewed based on titles and abstracts, 977 articles were deemed to be not relevant to the study and were removed (Figure 1), leaving a total of 59 articles. The 59 articles were subjected to a full-text review eligibility assessment and screened based on the inclusion and exclusion criteria of this study. At the end of the full-text review, only 19 articles were found to be eligible to be included in this review (Figure 1).

### 3.2. Molecular Mechanisms Modulated by LIPID-Soluble Vitamins in GBM

Out of the 19 eligible studies analyzed, seven papers (*n* = 7) reported on the effects of different forms of retinoids and vitamin A on human GBM cells (Table 1). There were six studies (*n* = 6) reporting on the effects on the active form of vitamin D3 (D3) and one study on the effects of vitamin K1 (VK1) on human GBM cells. Finally, of the five articles (=5) on the effects of vitamin E derivatives on human GBM cells, three (*n* = 3) were on tocopherols (Toc) and two (*n* = 2) were on tocotrienol (T3) (Table 1). The anti-tumoral effects exerted by these lipid-soluble vitamins on various types of GBM cell lines were observed to work through a diverse range of molecular mechanisms and cellular activities, such as cell differentiation, cell cycle arrest, apoptotic cell death, and other mechanisms that interfere with cell survival and proliferation as cell invasion, adhesion and migration, various transporters, and multidrug resistance proteins (Table 1).

#### 3.2.1. Retinoids (Vitamin A)

Retinoids constitute a large family of lipid-soluble vitamin A derivatives that include naturally occurring and synthetic molecules with similar properties. These compounds participate in various biological functions, such as embryogenic differentiation and growth, immunological activities, hematopoiesis, bone metabolism, cellular proliferation, and apoptosis. Retinols and their esterified derivatives, retinaldehyde and retinoic acid (RA), are naturally occurring vitamin A compounds. Natural retinoids are highly valued in cancer treatment due to their involvement in the regulatory effects of growth and differentiation at the cellular level [9]. Alterations in the normal homeostasis of vitamin A has been reported in many tumors, including breast, prostate, skin, leukemia, and cervical carcinomas [16,17]. Retinoids may affect tumorigenesis in GBM cells through several mechanisms (Figure 2). For instance, the modulation of the retinoid X receptor (RXR) was reported in three different GBM cell lines (U87MG, A172, and T98G) treated with retinols [18], which caused reduced the expression of proteins downstream of fatty acid (FA) and cholesterol synthesis (FAS and FDFT1, respectively). The epigenetic modification of the nervous system polycomb1 (NSPc1) gene was found to suppress the expression of retinol dehydrogenase-16 (RDH16), the enzyme that converts retinol to all-trans retinoic acid (ATRA) in GBM cells, resulted in the decreased synthesis of intracellular ATRA, which impeded cell differentiation and promoted development of stem-like cells (SLC) [19]. The treatment of SLC derived from two human GBM cell lines (U87MG and U251) with ATRA inhibited the expression of stemness markers such as CD133 and Sox2, and reversed the suppression caused by NSPc1 on RDH16, allowing cell differentiation to take place [19]. In another study, treating cancer stem cells (CSC) derived from the LN18 GBM cell line with ATRA, inhibited the expression of Nestin and induced cell differentiation through the activation of the extracellular signal-regulated kinase 1/2 (ERK1/2) [20].

**Figure 2 nutrients-14-02873-f002:**
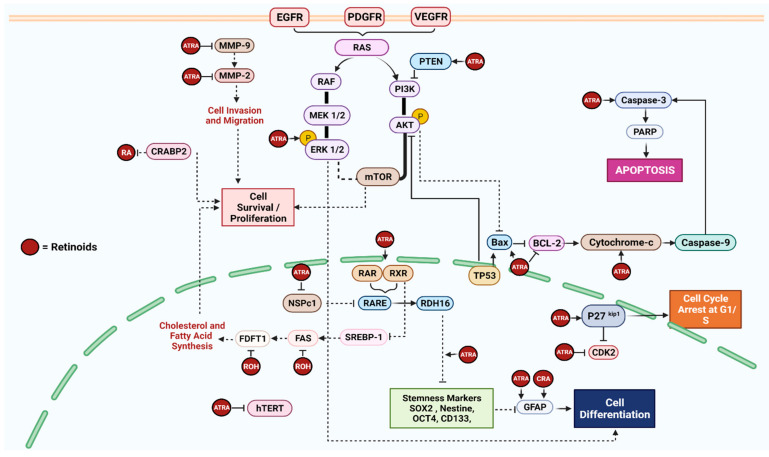
Illustration of the cellular pathways in GBM where the retinoids of vitamin A interact, identified from the systematic literature review of vitamin A [18,19,20,21,22,23,24]. (Created with BioRender.com).

**Table 1 nutrients-14-02873-t001:** Summary of outcomes induced by the lipid-soluble vitamins on various human glioblastoma cell lines.

Ref.	Aim of the Study	Glioblastoma Cell Line(s)	Vitamin	Intervention	Main Outcome	Other Outcomes
[18]	To test the effect of phytol and retinol on the viability of human GBM cell lines To determine the effect of phytol and retinol on the expression of signaling pathway and key proteins belong to cholesterol and/or fatty acid biosynthesis	U87MG, A172, and T98G	Retinol	U87MG-41.8 µM, A172–10.4 µM, and T98G-244.2 µM for 72 h	Transcriptome analysis revealed the downregulation of genes involved in the cholesterol and/or fatty acid biosynthetic pathway by retinol IC_50_ Retinol significantly reduced the level of (FAS and FDFT1) key proteins involved in tumor lipogenesis and cholesterol synthesis	Retinol exhibited a dose-dependent cytotoxic effect on GBM cell lines
[19]	To correlate the increased expression of NSPc1 and increase in tumor growth in stem-like cells (SLC) of GBM cell lines To test the effect of ATRA on SLCs and expression of NSPc1 and stemness markers	U87MG-SLC and U251-SLC	ATRA	10 μM (6–12 days)	ATRA treatment partially reversed NSPc1 induced-stemness markers (CD133 and Sox2), resulting in the ATRA-induced differentiation of GBM stem cells through the activation of the RDH16 protein	ATRA partially reversed glioma sphere growth in stem-like cancer cells and promoted differentiation in U87MG-SLC cells. NSPc1 knockdown resulted in impaired neurospheres’ formation, self-renewal abilities, and the downregulation of stemness markers CD133 and Sox2 NSPc1 epigenetically repressed the expression of RDH16 by directly binding to the RDH16 promoter
[20]	To investigate the role of mTOR in CSC maintenance and to establish the mechanism of targeting GBM CSCs using differentiating agents along with inhibitors of the mTOR pathway	U87MG and LN18	ATRA	10 μM for 2, 6, and 24 h	ATRA induced the differentiation of CSCs resulting in (1) suppression in the stem cell marker Nestin and (2) the enhanced expression of activated extracellular signal-regulated kinase 1/2 (pERK1/2) “independent of mTOR pathway inhibitors”	The combination of AT-RA, PI3K inhibitor, and mTOR inhibitor synergistically resulted in reduced CSC proliferation and migration
[21]	To demonstrate the presence of CRABP2 predominantly in the cytoplasm of GBM To correlate the relationship between CRABP2 level and RA treatment	U251 and M049	RA	0.5–5 µM for 6 h	Following RA treatment, CRABP2 accumulate in the cytoplasm of GBM cells, blocking the action of RA and activating anti-apoptotic pathway proteins (Cyclin E/CDK2, CRYAB, GFAP, and FABP7)	Knockdown of CRABP2 reduced proliferation rate, restored RA function, and downregulated the expression of anti-apoptotic proteins (Cyclin E, CRYAB, GFAP, and FABP7)
[22]	To investigate the effect of ATRA treatment on the migration, invasion, apoptosis, and proliferation of glioma cells	U87MG and SHG44	ATRA	5, 10, 20, or 40 µmol/L for 24 h	ATRA significantly downregulated the expression of invasion-mediated factors (MMP-2 and MMP-9) in a dose-dependent manner in all lineages. However, MMP-2 expression in U87MG cell line was only lowered following a high dose treatment of ATRA (20 and 40 µmol/L)	ATRA significantly inhibits the migration, invasion, proliferation, and promotes the apoptosis of GBM cells following treatment with various concentrations for 24 h in a dose-dependent manner
[23]	To investigate and compare the effect of ATRA and/or Interferon-γ on different GBM cell lines, LN18 (*PTEN*-proficient) and U87MG (*PTEN*-deficient)	LN18 and U87MG	ATRA	1 μM for 7 days	In LN18 cells, ATRA induced cell differentiation followed by the elevation of GFAP and apoptosis by increasing the Bax: Bcl-2 ratio, the mitochondrial release of cytochrome c into the cytosol, and calpain and caspase-3 activity, which is triggered by *PTEN* expression. While U87MG failed to show apoptotic biomarkers, it still exhibited high levels of GFAP, reflecting differentiation. ATRA also elevated p27^kip1^ and decreased CDK2 levels in both cell lines, indicating cell cycle arrest at the G_1_/S phase	The combination of ATRA and Interferon-γ control the growth of both GBM cell lines (*PTEN*-proficient and *PTEN*-deficient) by inducing cell differentiation, apoptosis, and cell cycle arrest
[24]	To test whether cell differentiation induced by the retinoids (ATRA or 13-CRA) affects GBM cells’ sensitivity to the microtubule-binding drug Taxol (TXL) and triggers apoptosis	T98G and U87MG	ATRA and 13 CRA	1 μM ATRA or 13-CRA for 7 days	GBM cells treated with ATRA or 13-CRA induced astrocytic differentiation, followed by the overexpression of GFAP and the downregulation of hTERT expression and activity, but no effects on apoptotic pathway proteins were found	The combination of retinoids with TXL effectively enhanced cell differentiation and apoptosis
[25]	To interrogate the possible functions of 1α,25(OH)2 vitamin D3 on mutant P53 and wild-type GBM cell lines	GL15 (wild-type *P53*), U251, and LN18 (mutant *P53*)	VD3	100 nM and 400 nM for 24 h	VD3 act via vitamin D receptor (VDR) in GL15 cells at a concentration of 100 nM as well as neutral sphingomyelinase1 by increasing the expression nSMase, aSMase, and GFAP, while increasing ceramide levels in U251 and LN18 cells at a concentration of 400 nM	The differentiation of mutant p53 (U251 and LN18) cells induced by neutral Sphingomyelinase1 enzyme was observed following a 24 h treatment with 400 nM (high dose) VD3 using the immunofluorescence method
[26]	To investigate the effects of 1α,25(OH)2 vitamin D3 on the expression of stemness markers in stem cell-like glioma cells in an acidic microenvironment	U87MG, T98G, and U251	VD3	10 nM and 100 nM in 4, 8, 12, 24, and 48 h intervals	VD3 10 nM or 100 nM treatment for 4–24 h suppressed the expression of stemness markers (Nestin, Oct4, and Sox2) on stem-like glioma cells	Acidosis induced the self-renewal ability of neurospheres, which were markedly reduced when treated with 10 or 100 nM VD3 under pH 7.4 and pH 6.8 conditions. VD3 inhibited the ATP production of mitochondria and rescued the acidosis triggered ATP enrichment
[27]	To investigate the effects of 1α,25(OH)2 vitamin D3 on the expression of senescence markers in glioma cells	U87MG and U251	VD3	10, 100, or 500 nM for 48 h	VD3 significantly increased the senescent markers INK4A (p16) and CDKN1A (p21) and promoted the expression of histone demethylase KDM6B glioma cells, while it does not affect vitamin D receptor expression	KDM6B knockdown attenuated VD3 and induced the senescence of glioma and reduced INK4A and CDKN1A upregulation
[28]	To investigate the expression of vitamin D receptor (VDR) in human glioma tissues To evaluate the effect of 1α,25(OH)2 vitamin D3 on cell survival and the modulation of the cell cycle in GBM cell lines	U251, U87MG, and T98G	VD3	1 μM for 96 h	VD3 reduced cell survival and induced cell cycle arrest witnessed by the increase in the expression of p57, p27, and p21 and a decrease in Cyclin D1 expression	An increase in the expression of VDR was denoted in human GBM cells as compared with the non-malignant control
[29]	To analyze five 1,25(OH)2 VD3-resistant GBM cell lines for key components of notch-signaling pathways using conventional RT-PCR	TX3868, U373, U118, TX3095, and U87	VD3	10−6 mol/L (4 h) 10−8 mol/L (24 h)	Treatment with various concentrations of VD3 failed to modulate the expression of any key component of the notch-signaling pathway	Combination treatment of VD3 with TSA or 5-aza did not enhance antiproliferative effect. but in fact reduced it, indicating a protective antagonizing effect of VD3 against the effect of other treatments
[30]	To detect the metabolism of vitamin D3 by tracking the expression of CYP27B1 splice variants To investigate the effect of vitamin D3 metabolites on GBM cell proliferation	TX3868 and TX3095	VD3	10–8 mol/L (24 h.)	VD3 increased the expression of CYP27B1, 1α25-dihydroxy vitamin D3-24 hydroxylase (CYP24), showing that GBM cell lines were able to metabolize VD3, while it showed no effect on the expression of VDR	VD3 metabolites increased the proliferation of GBM cell lines in a dose-dependent manner
[31]	To investigate the anticancer effects of vitamin C/E and Methotrexate on GBM	DBTRG	α-Toc	5 µM (24, 48, and 92 h)	α-Toc did not reveal any changes in the expression of proteins related to the caspase-3 death pathway (Cleaved PARP, Caspase-3, and Cleaved Caspase-3) on its own in the GBM cell line	The combination of vitamin E with low dose (0.01 µM) methotrexate displayed a significant anti-cancer effect on the GBM cell line through the activation of the Caspase-3 pathway
[32]	To apply combinatorial approach with the joint application of γ-Tocotrienol and jerantinine A to minimize toxicity towards non-cancerous cells and improve potency on brain cancer cells	U87MG	γ-T3	3.17 μg/mL (24 h)	Individual treatment with γ-T3 induced anti-proliferative effect on U87MG cells through multiple mechanisms: upregulation of pro-apoptotic protein Bax, TRAIL and Caspase-3 and Caspase-8 enzymatic activity, while inducing cell cycle arrest at G0/G1 phase and double-stranded breaks	The combined use of γ-T3 and jerantinine A induced the disruption of the microtubules network and Fas and p53 activation, triggering apoptosis. This demonstrated an improved potency of γ-T3-induced apoptosis through death receptor and mitochondrial pathways
[33]	To compare the cytotoxicity potency of alpha-, gamma-, and delta-tocotrienol and to explore the resultant apoptotic mechanism in glioblastoma U87MG cells	U87MG	α-, γ-, and δ-T3	24 h IC_50_ concentrations: α-T3 (2.0 µM), γ-T3 (3.0 µM), and δ-T3 (1.0 µM)	α-, γ-, and δ-T3 efficiently inhibited cell growth in a time- and concentration-dependent manner by triggering both intrinsic (Bax, Bid, and Cytochrome c), which was confirmed by a decrease in mitochondrial membrane permeability (MMP), and extrinsic (Caspase-8) pathways of apoptosis downstream signaling components	δ-T3 was found to be the most potent isomer among the tested isomers of tocotrienol
[34]	To evaluate the effect of γ- and α-tocopherol on the proliferation, integrin expression, adhesion, and migration of human GBM cells	U87MG	γ-Toc and α-Toc	50 μM for 6 h	Both γ- and α-toc increased the expression of integrin α5 and β1 protein and cell surface heterodimer integrin α5 β1, resulting in decreased proliferation and adhesion to fibronectin	γ-Toc exerted more anti-proliferative effect than α-Toc, while having a controversial impact on cell migration
[35]	To evaluate the effect of α-tocopheryl succinate on the expression of MRP1 and intracellular glutathione level in GMB when co-treated with VP-16	U87MG, T98G and U251MG, SNU 489	α-TOS	50, 100, and 150 μM	α-TOS decreased the expression of MDP1 and resulted in chemosensitization of GBM cells to VP-16	α-TOS decreased the intracellular concentration of glutathione
[36]	To characterize the efficacy and possible mechanisms of the combination of sorafenib and vitamin K1 on glioma cell lines	BT325 and U251	VK1	50 μM for 24 h	VK1 as an individual treatment failed to induce apoptosis or changes to the Raf/MEK/ERK signaling pathway	VK1 enhanced the cytotoxicity of sorafenib by repressing the Raf/MEK/ERK signaling pathway and inducing apoptosis in GBM cell lines

**13CRA:** 13-*cis* retinoic acid; **ATRA:** all-*trans* retinoic acid; **GBM:** glioblastoma; **α-T3:** α-Tocotrienol; **α-Toc:** Tocopherol; **δ-T3:** δ-Tocotrienol; **γ-Toc**: γ-Tocopherol; **γ-T3:** γ-Tocotrienol; **α-TOS:** α-Tocopheryl succinate; **RA**: retinoic acid; **VD3**: 1α,25(OH)2 vitamin D3; **VK1**: vitamin K1.

Cell invasion and migration were inhibited in GBM cells (U87MG and SHG44) treated with ATRA, due to the suppression of matrix metalloproteinase-2 (MMP2) and MMP-9 (Figure 2) [22]. In addition, LN18 cell exposure to ATRA was reported to upregulate the expression of apoptotic proteins, such as Bax:Bcl-2 ratio, cytosolic cytochrome C, and Caspase-3, which in turn promoted the expression of the tumor suppressor gene, PTEN [23] (Figure 2). However, changes in these apoptotic markers were not observed in U87G cells, which do not express PTEN. Nevertheless, ATRA caused cell cycle arrest at the G1/S phase through the upregulation of cyclin-dependent protein kinase inhibitor 1B (CDKN1B) “p27^kip1^” and the consequent suppression of CDK2 in U87MG cells [23]. In contrast, treating PTEN-deficient U87MG cells with 13-cis retinoic acid (13 CRA) induced cell differentiation due to the overexpression of the glial fibrillary protein (GFAP) and the downregulation of the human telomerase reverse transcriptase (hTERT), but did not have any effects on the proteins involved in the apoptotic pathway [24]. These findings suggest that ATRA may trigger apoptotic pathways in tumors that express PTEN.

#### 3.2.2. Vitamin D3

The active form of vitamin D3 (VD3), known as 1α,25(OH)2 vitamin D3, is a fat-soluble secosteroid hormone obtained from the diet or biosynthesized following a cascade of biochemical reactions. This micronutrient regulates a range of cellular physiological effects, such as cell proliferation, differentiation, angiogenesis, apoptosis, and metastasis [37]. In the brain, VD3 is reported to exert neuroprotective and neuro immune-modulatory functions [38]. In the biosynthesis of VD3, 7-dehydrocholesterol in the skin is converted to vitamin D post-exposure to ultraviolet (UV) light [30]. Then, the vitamin D undergoes hydroxylation in the liver to produce 25-hydroxy vitamin D, which is converted to the biologically active form 1α,25(OH)2 VD3 (also known as calcitriol) by the alpha-hydroxylase enzyme in the kidneys [37]. Calcitriol plays a significant role in calcium homeostasis and the development of the immune system and bones [9].

Human GBM cells treated with VD3 showed increased expression of the cyclin-dependent kinase inhibitors (CDNK1) CDKN1A (p21), CDKN1B (p27), and p57 and reduced expression of cyclin D1 (Figure 3), and the response was mediated through the vitamin D receptor (VDR), which resulted in cell growth arrest [28]. In another study, VD3 induced apoptosis, cell cycle arrest, and cell differentiation through the activation of neutral sphingomyelinase 1 (nSMase1), which degraded sphingomyelin and increased the ceramide pool in P53-mutant GBM cell lines (GBM U251 and LN18 cells) [25]. In addition to the ceramide pathway, VD3 was reported to increase the expression of senescence markers, such as cyclin-dependent kinase inhibitor 2A (CDKN2A or also known as INK4A or p16) and cyclin-dependent kinase inhibitor 1A (CDKN1A), due to the upregulation of lysine demethylase 6B (KDM6B) in U87MG and U251 GBM cells [27]. The authors proposed that the promoter of the *INK4* gene was demethylated at histone H3 (also known as H3K27me3), which is catalyzed by KDM6B, triggering senescence. Treatment with VD3 suppressed markers of stemness and self-renewal ability induced by an acidic microenvironment in stem cell-like (SCL) glioma cells, through the inhibition of ATP production and mitochondrial respiration [26]. The authors suggested that reduced acidosis following VD3 treatment may have downregulated the expression of the *CYP24A1* gene, which encodes for an enzyme responsible for the catalytic degradation of VD3 to prevent its cell modulation in these glioma cells. Some studies reported resistance to VD3 in several GBM cell lines, such as TX3868, U373, U118, TX3095, and U87 cells [29]. However, the TX3868 and TX3095 GBM cell lines showed dose-dependent proliferative responses post-VD3 treatment, which could be attributed to an increased expression of 1,25-dihydroxy vitamin D3-24 hydroxylase (CYP24) following exposure to VD3 [30], suggesting that although these GBM cells could metabolize VD3, it did not affect the expression of VDR.

#### 3.2.3. Vitamin E

Vitamin E is a hydrophobic, lipid-soluble antioxidant molecule with multiple health benefits. There are two major forms of vitamin E, known as tocopherol (Toc) and tocotrienol (T3), and each form exists in four natural isoforms, namely alpha (α), beta (β), delta (δ), and gamma (γ) [39], which means that there are eight naturally occurring derivatives of vitamin E. Vitamin E can only be obtained from the diet. The main sources of Toc are vegetable oils, such as wheat germ, canola, and sunflower, while T3s are present in high content in cereal grains, rice bran oil, palm oil, and annatto bean oil [40]. Vitamin E is well known for its antioxidant activities [41], and more recent work have highlighted the anti-cancer properties of vitamin E [42] (Figure 3).

**Figure 3 nutrients-14-02873-f003:**
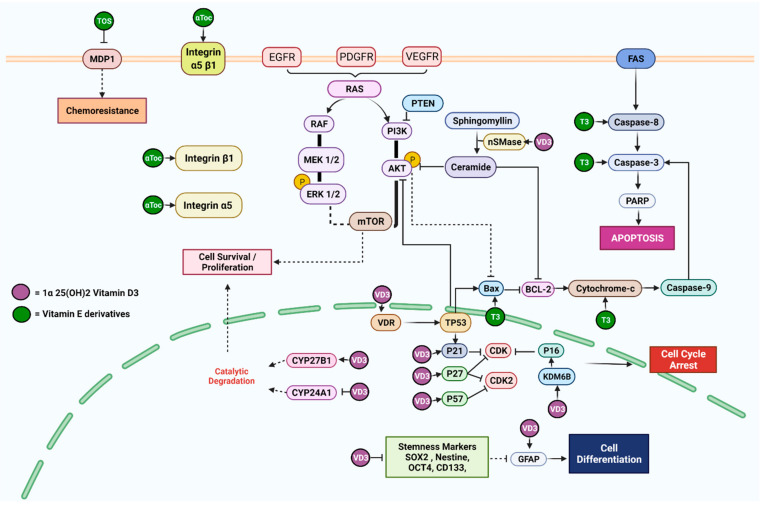
Illustration of the cellular pathways in GBM where vitamin D3 and vitamin E derivatives interact, identified from the systematic literature review of vitamin D3 [25,26,27,28,29,30] and vitamin E derivatives [31,32,33,34,35]. (Created with BioRender.com).

Alpha-Toc induced cell cycle arrest in cancer cells through the inhibition of protein kinase C (PKC) and reduced cyclin D1 and cyclin E levels [43]. Both α- and γ-Toc inhibited the proliferation of U87MG cells by inducing the expression of β1 and α5 proteins, which blocked cell cycle progression and the upregulation of the integrin α5β1 surface heterodimer that decreased cell adhesion to fibronectin, thus attenuating cell survival [34]. However, in the same study, it was shown that, when used independently, γ-Toc increased cell migration, while α-Toc failed to activate the apoptotic pathway in the GBM cells. However, there was a marked increase in cell death and the expression of various apoptotic biomarkers observed in GBM cells exposed to vitamin E and low doses of methotrexate [31]. In another study, exposure to α-Tocopheryl succinate (TOS), an esterified derivative of α-Toc, induced the chemosensitization of GBM cells to etoposide, triggering a reduction in intracellular glutathione concentration and blockage of multidrug resistance protein 1 (MDP1) [35]. These findings reflect the inconsistent anti-cancer effects of Toc.

Several studies have shown promising anti-cancer activities using different T3 isoforms. For instance, α-, γ-, and δ-T3 inhibited the proliferation of U87MG cells in a time- and dose-dependent manner, with a higher efficacy observed with δ-T3 [33]. The authors reported that the T3-treated GBM cells were arrested at the pre-G1 phase, which was accompanied by DNA damage and a concomitant increase in Caspase-8, BCL-2 Associated X (Bax), and BH3 interacting-domain death agonist (Bid) levels as well as the cytosolic release of cytochrome c [33]. These findings suggest that T3 might be involved in crosstalk between the intrinsic and extrinsic pathways of apoptosis in GBM cells. In another study, treatment with γ-T3 activated the extrinsic apoptotic pathway in brain cancer cells through the upregulation of pro-apoptotic proteins, such as Bax, and TNF-related apoptosis-inducing ligand (TRAIL), Caspase-3, and Caspase-8 also induced cell cycle arrest at G0/G1 phase [32]. When the brain cancer cells were exposed to γ-T3 and jerantinine A, there was a disruption of the microtubules network and induction of the mitochondrial pathway, and FS-7-associated surface antigen (Fas)- and p53-mediated apoptotic cell death [32].

#### 3.2.4. Vitamin K1

Vitamin K is a lipid-soluble vitamin that exists in three forms, two of which are natural forms, i.e., vitamin K1 (phylloquinone) obtained from green leafy vegetables and vitamin K2 (menaquinone), which is synthesized by the intestinal flora [44]. The anti-cancer properties of vitamin K have been tested on various cancer cell models, including colon, lung, liver, breast, stomach, leukemia, nasopharynx, and oral epidermoid cancers [9]. However, there is little literature describing the biomarkers modulated by the natural forms of vitamin K in GBM cells. In addition, vitamin K1 on its own did not induce any cytotoxic effects on the BT325 and U251 cell lines of GBM, but, when combined with sorafenib, marked synergistic anticancer effects were observed, which was shown to work via the suppression of the Raf/MEK/ERK signaling pathway that induces apoptosis [36].

### 3.3. Differentially Expressed and Replicable Biomarkers Modulated by Lipid-Soluble Vitamins in GBM

From the 19 eligible studies shortlisted for this review (Table 1), there were a total of 40 biomarkers that were differentially expressed in human GBM cells. Exposure to retinoids altered the expression of 21 biomarkers in GBM cells that were associated with apoptosis, cell differentiation, cell cycle arrest, and cell proliferation/survival (Figure 4). VD3 modulated the expression of 15 biomarkers involved in apoptosis, cell cycle arrest, and cell proliferation/survival pathways in GBM cells, while vitamin E modulated 11 biomarkers involved in apoptosis, chemoresistance, and cell proliferation/survival cellular signaling pathways (Figure 4). Twelve of these biomarkers were reported in two or more studies within the same vitamin group or between different vitamins. Four biomarkers (GFAP, Nestin, p27, and Sox2) were found to be common between retinoids and vitamin D3 with respect to human GBM cells. In particular, GFAP was reported in four independent studies (Table 1), with three studies using retinoids and one study using VD3 [21,23,24,25]. In addition, the expression of three biomarkers related to apoptosis (Bax, cytochrome c, and Caspase-3) was altered in GBM cells treated with retinoids [23] or T3 (vitamin E) [32,33] (Figure 4). However, there were no common biomarkers that were differentially regulated by VD3 or vitamin E derivatives in GBM cells. Two studies reported that cyclin-dependent kinase 2 (CDK2/Cyclin E), a protein kinase that regulates the cell cycle transition from G1 to S, was differentially regulated by retinoid in GBM cells [21,23]. Two independent studies reported that vitamin E affected the expression of the apoptotic biomarker Caspase-8 in GBM cells [32,33].

### 3.4. Clinical Survival Correlation of Biomarkers

The clinical survival correlation of six biomarkers (*CDK2*, *GFAP*, *CDKN1B*, *CDKN1A*, *BAX*, and *CASP8*) identified in this systematic descriptive review was analyzed using datasets in The Cancer Genome Atlas (TCGA). These biomarkers were chosen for the clinical correlation analysis as these were reported in two or more of the shortlisted studies included in this review. In addition, these six biomarkers were consistently modulated by vitamins A, D, or E in two or more studies. From the literature, these genes (*CDK2*, *GFAP*, *CDKN1B*, *CDKN1A*, *BAX*, and *CASP8*) encode for CDK2 [21,23], GFAP [21,23,24,25], p27 [23,28], Bax [23,32,33], and Caspase-8 [32,33] proteins.

The Kaplan–Meier analysis revealed that a higher expression of *CDK2* (Figure 5a), *BAX* (Figure 5e), and *CASP8* (Figure 5f) genes is associated with a decreased median survival in GBM patients, which suggests that these genes/proteins may have negative prognostic values in GBM. However, only *BAX* showed significant (*p* = 0.033) mean survival values. In contrast, the increased expression of *GFAP* (Figure 5b), *CDKN1B* (Figure 5c), and *CDKN1A* (Figure 5d) genes appears to correspond with an increased overall median survival, suggesting that these may have a positive prognostic value. However, only *CDKN1B* showed significant (*p* = 0.039) mean survival values.

## 4. Discussion

Several core pathways have been implicated in the carcinogenesis of brain cancers, including those associated with growth regulation, DNA repair, cell cycle regulators, apoptotic cell death, cell invasion, adhesion, migration, and angiogenesis [8]. In addition to genetic heterogeneity, GBM tumors exhibit functional heterogeneity, causing the hierarchical organization of the cells and the involvement of a small population of CSCs within the core of the tumor tissue, which allows the tumor to re-emerge [5]. The CSCs can migrate to adjacent tissues and exhibit biological changes that contribute to their survival and drug resistance as these cells are robust and quiescent in nature [8]. In the present study, we investigated how lipid-soluble vitamins may affect the development of GBM by evaluating the expression of biomarkers involved in various regulatory pathways (Figure 2 and Figure 3).

The current review investigated the role of 40 differentially expressed biomarkers, with 12 biomarkers being replicable, i.e., reported as differentially regulated in at least two independent studies. The activation of GFAP by retinoids and vitamin D3 is a good indicator of their potential initiation of cell differentiation that would ultimately impede the pro-proliferative machinery in GBM cells. In addition, cell differentiation is also linked with the increased activation of ERK1/2 following treatment with ATRA. The ERK1/2 is the downstream signal of the Ras/Raf/MEK1/2 pathway, which is altered in several cancers, including GBM. Activated ERK1/2 can induce the expression of several transcription factors that trigger multiple cellular processes, including uncontrolled cell proliferation, apoptosis resistance, and cell differentiation [45].

In general, the overexpression of GFAP and the suppression of biomarkers associated with stemness are investigated in most studies involving lipid-soluble vitamins and GBM. The analysis of this study showed that the expression of stemness markers, such as CD133, Oct4, Nestin, and Sox2, was inhibited in GBM cells treated with retinoids, while the expression of Nestin and Sox2 was suppressed in VD3-treated GBM cells. The findings of this study reflect the huge potential of retinoids and VD3 to induce cell differentiation and inhibit stemness in GBM cells, which suggest that these lipid-soluble vitamins may play a role in reducing drug resistance.

Apart from promoting cell differentiation and suppressing stemness, retinoids and VD3 promote cell cycle arrest/senescence by modulating various cyclin-dependent protein kinase inhibitors, such as p16(INK4A), p21(CDKN1A), p27 (CDKN1B), and p57, which was apparent in a PTEN-deficient cell-based model of GBM with ATRA [23]. Other effects include the suppression of human telomerase reverse transcriptase (hTERT), a protein that contributes to DNA replication, which was also marked on the latter model [24].

Conversely, the treatment with retinoic acid (RA) failed to cease growth and proliferation in GBM cells [21]. This was justified by the accumulation of the transporter protein CRABP2 in the cytoplasm of GBM cells. The main function of retinoids is utilized following its transport to the nucleus, where it binds to a ligand-based transcription-mediated receptor, retinoic acid receptor (RAR) and RXR [21]. Thus, the accumulation of CRABP2 in the cytoplasm contributed to retinoids’ transport failure.

Cell senescence was manifested in GBM cells exposed to VD3, which was independent of the vitamin D receptor (VDR) expression [30]. Moreover, the overexpression of histone demethylase (KDM6B) is cited as being a huge contributor to senescence induced by VD3 through the demethylation of the promoter of the *INK4A* gene [27], which suggests that growth arrest is a preferential mechanism caused by VD3 in several GBM cell lines.

The activation of apoptotic signaling pathways was observed in GBM cells treated with ATRA or T3 isoforms. The treatment of PTEN-proficient GBM cells (LN18) with ATRA modulated the expression of several biomarkers associated with apoptosis, including the Bax: Bcl-2 ratio, cytosolic release of cytochrome C, and the activation of Caspase-3 [23]. It appears that exposure to ATRA induced different anticancer effects in different GBM cells. For instance, in the PTEN-proficient cells, treatment with ATRA preferentially triggered apoptosis, but in PTEN-deficient cells, ATRA induced cell cycle arrest at the G1/S phase [23]. These findings highlight the multi-target potential of ATRA as an anticancer agent in GBM, which is not dependent on the tumor genetic heterogeneity. In contrast, the T3 isomers (α-, γ-, and γ-T3) predominantly activate proteins involved in apoptotic signaling, such as Bax, Bid, Caspase-3, Caspase-8, cytochrome c, and TRAIL, while suppressing the Bcl-2 ratio [32,33], which is reported in various cancers [46]. However, further studies are required to elucidate the molecular biomarkers utilized by T3 isoforms in GBM models as an insufficient number of studies have been reported to date.

Modulation biomarkers involved in cell proliferation patterns are evident in several studies with retinoids, VD3, and vitamin E. For instance, the suppression of fatty acid and cholesterol biosynthesis proteins, FAS and FDFT1, respectively, is observed in ATRA-treated GBM cells [18]. Metabolic remodeling, such as reprogramming cholesterol and fatty acid synthesis patterns, is highly apparent in GBM due to a high demand for growth and energy and multiple mutations on tumor suppressor genes and oncogenes [47]. In addition to metabolic changes, ATRA demonstrated a marked suppression of proteins that induce cell invasion and migration, namely metalloproteinase 2 and 9 (MMP-2 and MMP-9) [22]. A relatively similar process was seen with α- and γ- tocopherols (Toc) of vitamin E, where α- and γ-Toc increase the expression of integrin β1, integrin α5 proteins, and integrin α5β1 surface heterodimer, thus contributing to decreased cell adhesion to fibronectin and the attenuation of cell survival [34]. On the other hand, as observed in some GBM cell models, the catalytic degradation of vitamin D3 through the activation of CYP24A1 and CYP27B1 hampers the anti-tumoral effects of vitamin D3 and, hence, contributes to cell proliferation [30]. In fact, another study revealed the downregulation of CYP24A1 when vitamin D3 was treated in an acidic microenvironment where vitamin D3 contributed to the suppression of CSCs [26]. Vitamin E derivatives have displayed a unique chemosensitization feature. In particular, a single study on the esterified α-Toc, α-Tocopheryl succinate (TOS), demonstrated a decreased expression of multidrug resistance protein 1 (MDP1) accompanied by the intracellular reduction in glutathione concentration in GBM cells [35]. Such a feature has only been seen on studies on vitamin E and could open doors on the potential of vitamin E derivatives to overcome the issue of chemoresistance that mitigated the productive success of chemotherapeutic drugs.

Regarding the survival analysis, although the six replicable biomarkers hold positive and negative clinical correlation, most of the biomarkers failed to show a statistically significant change, except for *Bax* and *CDKN1B*. Despite its lower gene expression in GBM patients’ survival analysis, Bax is an important apoptotic biomarker seen in GBM cell lines modulated by both ATRA and T3 isoforms. On the other hand, a higher expression level of *CDKN1B* is remarkably a positive prognostic marker and has contributed to the increased median survival of patients from the cohort survival analysis. ATRA and vitamin D3 regulate CDKN1B. Since GBM TCGA expression datasets represent a large-scale based set of information on GBM heterogeneity and microenvironment [48], the further segmentation of the patient population using GBM molecular subtype markers might result in a different overall survival outcome and provide a more detailed picture on the regulation of GBM biomarkers. Overall, little is known about the biomarkers regulated by the lipid-soluble vitamins in GBM, and further studies are required to structure their anti-tumoral role.

## 5. Conclusions

In recent decades, the elucidation of the anti-cancer potential of lipid-soluble vitamins (retinoids, vitamin D3, and vitamin E) has attracted researchers’ interest. The anti-cancer effects of lipid-soluble vitamins in various GBM cell lines are attributed to a diverse range of molecular mechanisms through changes in the expression of a wide range of biomarkers, of which *CDK2*, *GFAP*, *p27*, *Bax*, and *CASP8* genes may have some clinical value. Despite the undeniable outcome that has been achieved to date, the molecular mechanisms exerted by some of the vitamins remain poorly understood. Therefore, further studies are required to investigate the role of lipid-soluble vitamins in chemosensitization when combined with chemotherapy, including the various altered biomarkers and their possibility to prevent the spread of GBM.

## Figures and Tables

**Figure 1 nutrients-14-02873-f001:**
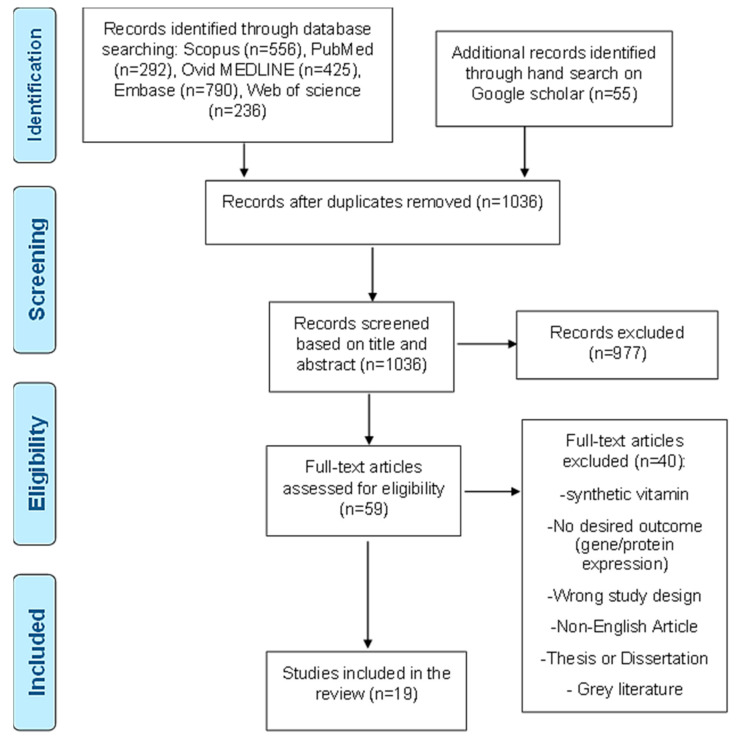
Systematic review PRISMA chart showing the literature search from 2005 to June 2021.

**Figure 4 nutrients-14-02873-f004:**
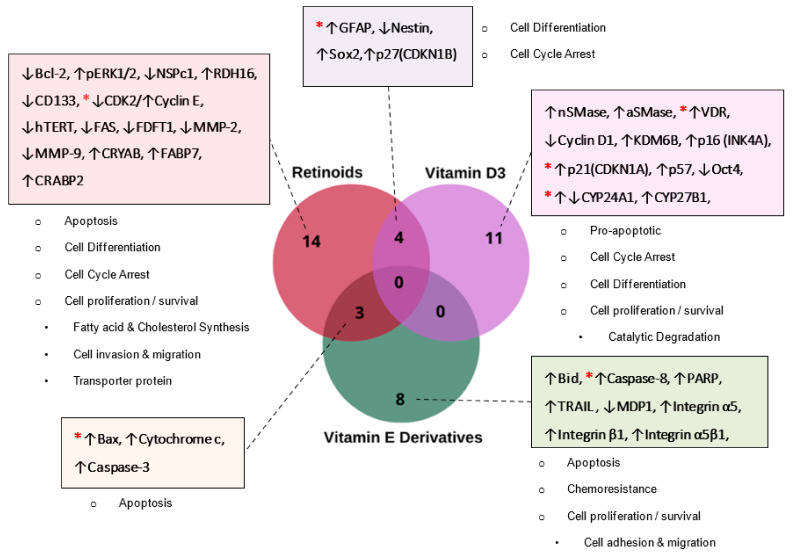
Venn diagram showing biomarkers regulated by the lipid-soluble vitamins A ‘Retinoids’ (red circle), D3 (purple circle), and E derivatives (green circle), with the corresponding molecular mechanism identified from the 19 eligible studies shortlisted in Table 1. ***** Represents a replicable biomarker within the same vitamin group.

**Figure 5 nutrients-14-02873-f005:**
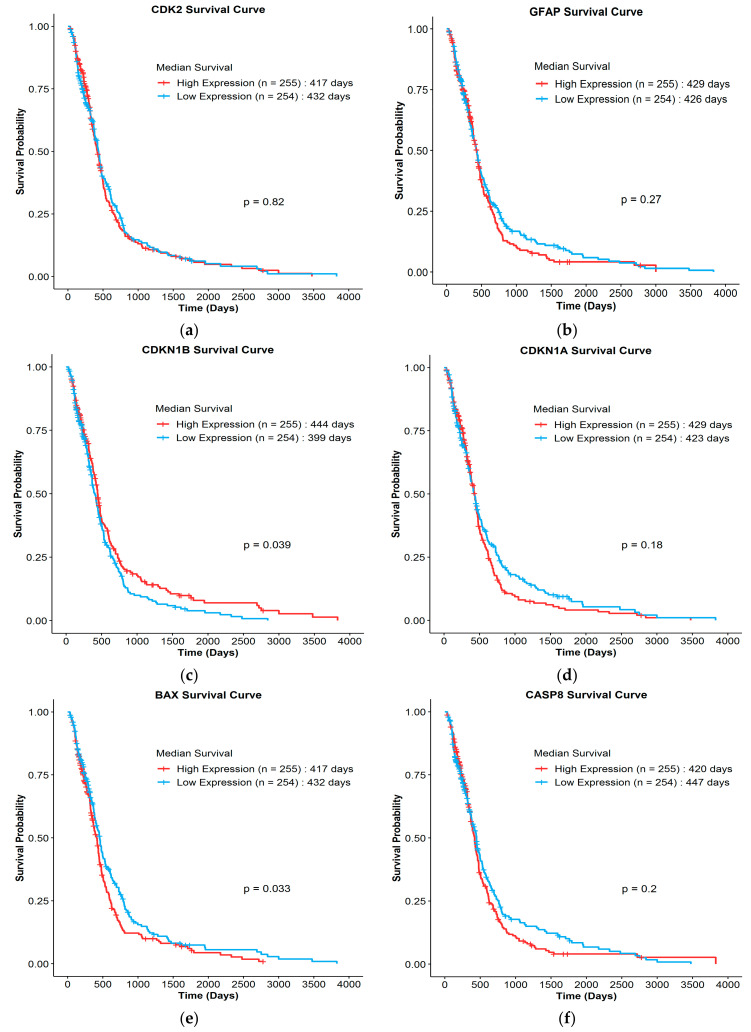
Kaplan–Meier survival analysis plot of 509 GBM patients from TCGA with an overall survival of more than 30 days based on the expression of genes correlated to the replicable biomarkers: (**a**) *CDK2*, (**b**) *GFAP*, (**c**) *CDKN1B* (*P27*), (**d**) *CDKN1A* (*P21*), (**e**) *BAX*, and (**f**) *CASP8*. Patients were grouped based on the median expression patterns into high and low expression groups using the log-rank test (a *p*-value of <0.05 indicates a statistically significant results).

## Data Availability

Not applicable.

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
