# Peer review of "Biomarkers Regulated by Lipid-Soluble Vitamins in Glioblastoma"

_nutrients, 2022, doi:10.3390/nu14142873_

Round 1

Reviewer 1 Report

A brief summary:

The authors present a review based on a systematic analysis of the literature searching for GBM biomarkers regulated by lipid-soluble vitamins. The analysis results in a detailed summary of 19 relevant research articles from 2005 to 2021 (with almost half within the last 5 years), which offers a general picture of the current knowledge regarding the biological impact of lipo-soluble vitamins in GBC cell signaling. The authors identify 12 proteins from the bibliographic analysis as biomarkers with positive or negative interaction with at least one of the studied vitamins. Six of these biomarkers coded by the genes (CDK2, GFAP, 353 CDKN1B, CDKN1A, BAX, and CASP8) were consistently modulated by vitamins A, D, or E in two or more studies and therefore considered to have a potential impact in GBC progression. The latter is evaluated in a GBC cohort from a TCGA gene expression dataset identifying Bax and CDKN1B levels as statistically relevant for the clinical outcome.

General concept comments:

  • Is the review clear, comprehensive and of relevance to the field? – The review is clear and comprehensive. The systematic analysis used offers a general picture of the current knowledge regarding the biological impact of lipo-soluble vitamins in GBC cell signaling.
  • Was a similar review published recently and, if yes, is this current review still relevant and of interest to the scientific community? – The information compiled and presented in the review is of interest to the scientific community.
  • Are the cited references mostly recent publications (within the last 5 years) and relevant? Are any relevant citations omitted? – The review presents a systematic summary of 19 relevant research articles from 2005 to 2021(with almost half within the last 5 years) in the cellular and molecular context of human GBC.
    Does it include an excessive number of self-citations?
    – The review does not include excessive self-citation.
  • Are the statements and conclusions drawn coherent and supported by the listed citations? -Yes.
  • Are the figures/tables/images/schemes appropriate? Do they properly show the data? Are they easy to interpret and understand? – Yes, but some figures need to be a bit improved (see specific comments).

Specific comments:

The authors are encouraged to upgrade or further explain the following comments:

·        Line 39: Specify the information of the three subtypes.

·        Line 48: The use of 5-Aminolevulinic Acid for maximal surgical resection aimed to help in the identification of tumor borders is common nowadays. DOI: 10.1227/NEU.0000000000000929

·        Line 66: Comment on the special case of vitamin D biosynthesis: cholesterol and sun exposure (UVB radiation). DOI: 10.1016/j.chembiol.2013.12.016, DOI: 10.3390/cells10082007.

·        Line 171. Correct the abbreviation for fatty acid synthase for FAS or FASN.

·        In table 2:

Correct the abbreviation for fatty acid synthase. Move reference [20] up to follow the same order of appearance as in the text.

o   Suggestion: Table 2 could be also improved by adding the DOI Link to the Reference number.

·        Figure 2. Correct Fatty acid synthase abbreviation.

·        Line 293. Specify the gene name of Fas.

·        Figure 4 should be improved:

o   Biomarkers could be organized according to the effect of lipid-soluble vitamins in each of the Venn diagram groups.

o   Image resolution should be improved.

·        All figures' should present a brief description and include references to review tables whenever it is convenient.

·        Line 364-5: Figure 5 gene symbols should be in italic or specified as gene expression.

·        Line 366: Figure 5 description should include detailed information regarding the database and dataset used and the applied statistical analysis.

·        Lines 451-459: discussion should also take into account other limitations of the survival analysis performed. GBM TCGA gene expression datasets come from bulk analysis of biopsies which contain information on GBM heterogeneity and microenvironment. Further stratification of the patient cohort using the GBM molecular subtype signatures (DOI: 10.1038/s41598-019-43173-y) could result in different overall survival outcomes and give more specific details regarding the regulation of the mentioned biomarkers in GBM.

Author Response

Dear Reviewer 1,

Thank you for your valuable suggestions. We have made the necessary changes according to your suggestions. Please find the attached file for the responses to your comments.

Reviewer 2 Report

The presented work by Osman et al is well-written, and makes for an enjoyable, cohesive read. I only have a few minor comments/spell-checks that I would for the authors to address before I can make a recommendation for acceptance. These comments are mentioned below:

1) Line 15 : Cancer potential (not potentials)

2) Line 21 and 27: Please be consistent. Either write 40 or forty, and have all other number references mentioned either in alphabet or numerals. Change throughout the paper.

3) Lines 37-39 : Break sentence into two sentences for ease of reading. 

4) Line 39: What is IDH? Introduce the abbreviation.

5) Line 61: Remove 'such'. Antioxidants and co-enzymes are not cellular functions. 

6) Line 64: Vitamins ARE classified as ...

7) The introduction is missing key gaps as to why this work is of importance and why this review was conducted. Please address this.

8) Lines 122/130/136: Spelling errors. Please correct.

9) Table 1 and Fig. 1 are both repetitive and superfluous, as this information is already presented in the text. Either choose the table/figure or the text, not both.

Author Response

Dear Reviewer 2,

Thank you for your valuable suggestions. We have made the necessary changes according to your suggestions. Please find the attached file for the responses to your comments.
